# Next Steps for Immunotherapy in Glioblastoma

**DOI:** 10.3390/cancers14164023

**Published:** 2022-08-20

**Authors:** Toni Q. Cao, Derek A. Wainwright, Catalina Lee-Chang, Jason Miska, Adam M. Sonabend, Amy B. Heimberger, Rimas V. Lukas

**Affiliations:** 1Department of Neurology, Northwestern University, Chicago, IL 60611, USA; 2Department of Neurological Surgery, Northwestern University, Chicago, IL 60611, USA; 3Lou & Jean Malnati Brain Tumor Institute, Chicago, IL 60611, USA; 4Department of Medicine, Division of Hematology/Oncology, Northwestern University, Chicago, IL 60611, USA; 5Department of Neuroscience, Northwestern University, Chicago, IL 60611, USA; 6Department of Microbiology-Immunology, Northwestern University, Chicago, IL 60611, USA

**Keywords:** immunotherapy, glioblastoma, immune system

## Abstract

**Simple Summary:**

Prognosis for glioblastoma patients remains poor despite the current standard of care treatments. More recent investigations have focused on immunotherapy, which utilizes a patient’s immune system to target cancer cells. Though proven to be successful in non-central nervous system cancers, immunotherapies have yielded disappointing results for glioblastoma thus far. A variety of factors play into the efficacy of immunotherapy for glioblastoma and have become new areas of interest. Here we review both historical and emerging immunotherapeutic approaches, as well as the molecular factors that have been shown to impact the efficacy of immunotherapies.

**Abstract:**

Outcomes for glioblastoma (GBM) patients undergoing standard of care treatment remain poor. Here we discuss the portfolio of previously investigated immunotherapies for glioblastoma, including vaccine therapy and checkpoint inhibitors, as well as novel emerging therapeutic approaches. In addition, we explore the factors that potentially influence response to immunotherapy, which should be considered in future research aimed at improving immunotherapy efficacy.

## 1. Introduction

Glioblastoma is a primary brain tumor for which there is a substantial unmet need with respect to therapeutic options. Contemporary standard treatment for newly diagnosed disease involves maximum feasible surgical resection, radiation therapy, systemic therapy with the DNA-alkylating agent temozolomide, and regional therapy with alternating electrical fields [1]. For progressive disease, optimal management is less clear [2]. Immunotherapeutic approaches, including vaccines, immune checkpoint inhibitors, or cell-based therapies, have proven remarkably successful in a host of non-central nervous system (CNS) malignancies [3]. Unfortunately, such gains in durable survival have yet to be observed across unselected patient populations with glioblastoma in the phase 3 trials thus far conducted [4,5,6]. A number of factors, both known and unknown, contribute to the lack of benefit from immunotherapy observed thus far. In this review, we will examine recent clinical investigations that may shed light on these resistance mechanisms. Additionally, we will identify future approaches that may prove of value in utilizing immunotherapies to treat glioblastoma.

## 2. Established Immunotherapeutic Approaches: Enhancing T-Cell Activities

Glioblastomas have been shown to promote immunosuppression intratumorally as well as systemically [7,8]. Decreased systemic immune activity is associated with tumor progression and shorter survival times [9,10,11,12]. These findings suggest that immunotherapeutic approaches to enhance immune activity and/or minimize immunosuppression may prove beneficial in treating patients with these tumors. Two of the most extensively studied approaches are vaccine therapies and immune checkpoint inhibitors, as detailed below. Chimeric antigen receptor (CAR) T-cell therapy, which involves the administration of genetically modifying cytotoxic T lymphocytes expressing chimeric antigen receptors for tumor antigens, is reviewed separately [13].

### 2.1. Vaccine Therapies

Vaccine-based immunotherapy is designed to induce a specific immune response against tumor antigens. Vaccines can be categorized into “off-the-shelf” peptide vaccines targeting pre-specified epitopes or vaccines based on autologous patient tumor tissue [14]. Summaries of both completed and ongoing peptide and autologous vaccine therapies are listed in Table 1.

#### 2.1.1. Peptide Vaccines

EGFRvIII: The epidermal growth factor receptor (EGFR) deletion mutation, EGFRvIII, is expressed in about ~33% of GBM and promotes tumorigenesis through various mechanisms [15,16,17,18,19,20,21,22,23]. Signaling pathways induced by EGFRvIII activation are illustrated in Figure 1. The most notable peptide vaccine is rindopepimut, previously known as CDX-110, which targets EGFRvIII. This vaccine was studied in several clinical trials [24,25,26]. In the phase 3 ACT IV trial (NCT01480479), the addition of rindopepimut to standard radiation and temozolomide treatment in EGFRvIII-positive glioblastoma patients did not improve survival as compared to radiation and temozolomide alone (20.1 months vs. 20 months, HR 1.01) [27]. Furthermore, there was no significant difference in progression-free survival, tumor response, or quality of life measures in the rindopepimut group compared to the control group [27]. Interestingly, the randomized phase 2 ReACT trial (NCT01498328) demonstrated that the addition of rindopepimut to standard bevacizumab in relapsed, EGFRvIII-positive glioblastoma patients trended toward improved progression-free survival (PFS) at 6 months. While multiple endpoints were positive (overall survival (OS), overall response rate, and median duration of response), the study was overall limited by a small sample size (n = 73 patients) [6].

It should be noted that almost 60% of the patients in the ACT IV trial who were treated with rindopepimut and studied at tumor recurrence (21 of 37 patients evaluated of the 745 patients enrolled) subsequently lost EGFRvIII expression [27]. Loss of EGFRvIII has been described in additional studies [25,26,28]. However, EGFRvIII loss is not limited to those treated with rindopepimut and can still occur in nearly 50% of patients with EGFRvIII expressing glioblastoma as part of the natural evolution of glioblastoma [29].

EGFRvIII (epidermal growth factor receptor variant III) activates multiple signaling pathways for tumorigenesis, including MAPK, mTOR, and STAT pathways. 

IDH1: Isocitrate dehydrogenase (IDH) 1 mutations are a defining characteristic of the histologically similar tumor, the astrocytoma IDH mutated WHO grade 4, which was previously characterized as a glioblastoma [30,31]. These tumors were previously included in many trials for glioblastoma. Recent attempts have been made to create a peptide vaccine targeting the IDH1 R132H neoantigen. One recent phase 1 clinical trial studied the efficacy of the IDH1 R132H mutant peptide vaccine (NOA-16) in IDH1 mutated grade III–IV gliomas, which led to both mutation-specific T cell and humoral immune responses [32] (NCT02454634).

WT1: A number of other antigen peptide-based vaccines are also being explored in GBM. Wilms tumor 1 (WT1) was previously identified as a tumor-associated antigen (TAA) in high-grade glioblastoma [33]. Subsequent clinical trials were performed in humans, demonstrating the safety and tolerability of the WT1 peptide vaccine [34]. The resulting data demonstrated the induction of WT1-specific IgG antibodies as well as the identification of a potential biomarker [35,36]. This is still an actively researched approach [37,38].

CMV: Efforts in developing a cytomegalovirus (CMV) peptide vaccine began after one study found CMV immune reactivity in 100% of GBM patients, albeit with some inconsistent results [39,40,41,42]. The vaccine is aimed at pp6537, which is a major structural protein of CMV. Two strategies for targeting CMV have been evaluated in GBM: a pp65-pulsed dendritic cell (DC) platform [43] and a CMV-specific T-cell expansion approach [44,45]. These approaches have been shown to be safe, and early small studies have revealed promising results. Two recent sequential clinical trials (ATTAC and its expanded cohort trial ATTAC-GM, NCT00639639) using CMV pp65 DC vaccines in newly diagnosed GBM patients consistently demonstrated an improved OS, with the latter trial demonstrating mPFS of 25.3 mo and OS of 41.1 mo [45,46]. A third larger confirmatory trial, ELEVATE (NCT02366728), involving 43 patients who received the CMV DC vaccine after pretreatment with tetanus diptheria toxoid (Td), demonstrated a 3-year OS of 34% [46,47]. Though these results are encouraging, larger confirmatory studies are needed. 

Survivin: Survivin is part of the inhibitor of apoptosis (IAP) family of proteins and functions to regulate cell division and programmed cell death [48]. There have also been ongoing efforts using a survivin peptide mimic (SurVaxM) in GBM [49]. This vaccine is currently being evaluated in one phase 2 trial for newly diagnosed GBM (NCT02455557). It is also being studied in combination with pembrolizumab in recurrent GBM in a separate trial (NCT04013672) [50,51]. While the vaccine is well tolerated, efficacy is unknown.

TERT: TERT (telomerase reverse transcriptase) mutations occur in many GBM patients [52,53]. As a result, a recent vaccine strategy was developed to target the telomerase reverse transcriptase (TERT) protein [53]. The TERT peptide-based vaccine (UCPVax-Glio) was recently developed by Invectys in France and is currently being studied in both phase 1 and 2 trials (NCT04280848).

Multipeptide vaccines: Considering the very real possibility of antigen escape using a single peptide targeting vaccine, the latest class of peptide vaccines were engineered toward multiple tumor antigens simultaneously. An ongoing phase 1/2 study found TAS0313, a vaccine comprised of 12 Cytotoxic T-cell epitopes against 8 TAAs (derived from EGFR, KUA, LCK, MRP3, PTHRP, SART2, SART3, and WHSC2), to be safe and potentially efficacious [54]. In another recent phase 2 study, administration of the ICT-107 vaccine, comprised of 6 TAA (MAGE-1, HER-2, AIM-2, TRP-2, gp100, and IL13Rα2) pulsed into DC, led to a notable increase in PFS in GBM patients [55]. A third vaccine called EO2401, which targets three bacterial peptides that mimic tumor antigens (termed Oncomimics), is being explored in the context of nivolumab for progressive GBM (NCT04116658) [56]. Interim results are expected soon. While no conventional peptide vaccine platform has led to any change in the standard of care (SOC) of GBM, the results of these studies are eagerly anticipated.

Neoantigens: Cancer neoantigens are T-cell epitopes that exclusively arise from tumor-specific somatic DNA mutations and are subsequently highly immunogenic given the lack of expression in normal tissues [57]. Neoantigen vaccines have been studied previously in patients with melanoma [58,59,60] but have only recently been studied in the GBM population. A recent phase 1/1b trial (NCT02287428) demonstrated that administration of a personalized neoantigen vaccine (NeoVax) in new GBM patients induced circulating neoantigen-specific T-cell responses and resulted in intratumoral T-cell infiltration, a subset of which were neoantigen-specific [61]. Of note, T-cell response was limited to patients who did not receive concomitant steroid treatment with dexamethasone. Additionally, despite one patient having a large number of unique CD4+ and CD8+ T-cell clonotypes, the actual number of clonotypes identical to those from the relapsed tumor was significantly lower [61].

#### 2.1.2. Autologous Vaccines

A summary of clinical trials investigating autologous vaccines for glioblastoma is listed in Table 2. There have been several studies evaluating autologous vaccines in the context of dendritic cells (DCs) [62,63]. DCs are key players in immunosurveillance and immune regulation that present phagocytosed antigens to naïve T cells. DCVax-L is the most well-studied DC vaccine and is comprised of autologous DCs pulsed with autologous tumor lysate. The phase 3 DCVax trial (NCT00045968) investigated the impact of DCVax-L in addition to SOC (surgical resection and chemoradiotherapy) for newly diagnosed GBM patients. Due to the cross-over study design, nearly 90% of the patients in this study received DCVax-L, making an interpretation of the survival outcome challenging. In the overall intent to treat the patient population (investigational arm + control arm), the mOS was estimated to be 23.1 months. This study has benchmarked these findings to historical controls but has not yet evaluated its primary endpoint of PFS [62]. A recent phase 1 trial focused on an autologous DC vaccine pulsed with GBM stem-like cell line lysate, which was shown to be tolerable (NCT02010606). Further results are pending [63].

Heat shock protein (HSP) vaccines are designed to promote a specific antitumor inflammatory response [64]. HSPs operate as intracellular chaperones and can deliver tumor antigens to antigen-presenting cells (APCs), including DCs, to facilitate T cell-mediated cytotoxic death [64]. The most well-studied HSP vaccine is the autologous HSPPC-96 (Prophage) vaccine, which is comprised of patient-specific tumor antigens conjugated to HSP gp-96. A single-arm, phase 2 trial enrolled 41 recurrent GBM patients treated with the HSPPC-96 vaccine (NCT00293423). Results revealed an mOS of 42.6 weeks and mPFS of 19.1 weeks [65]. Notably, these outcomes were only compared to historical controls. A subsequent multi-arm phase 2 clinical trial investigated both HSPPC-96 and bevacizumab, though it was then closed to accrual early after interim analysis demonstrated no improvement in OS [66].

Of note, several of these clinical trials utilized historical controls as the active comparator. Many have deemed the use of historical controls as suboptimal due to potential differences in patient, disease, and therapeutic characteristics that subsequently confound results. Furthermore, historical controls face different survival trends when matched to contemporary patients, likely as a result of evolving therapeutic options [67,68,69]. Moving forward, the reliability of historical controls should continue to be kept in mind when designing clinical trials.

Other personalized vaccines, such as CeGaT, are commercially available. To the authors’ knowledge, there have been no peer-reviewed preclinical or clinical studies of CeGaT for glioblastoma.

### 2.2. PD-1 and PD-L1 Blockade

Immune checkpoints regulate the immune system as “gatekeepers” of immune responses to maintain self-tolerance [70]. These immune checkpoints can be stimulated and utilized by tumors, including glioblastomas, to evade the immune system. One of the most well-studied immune checkpoints and therapeutic targets is the PD-1/PD-L1 axis. Glioblastoma tumor cells express programmed death ligand 1 (PD-L1), which binds to PD-1 on T cells (see Figure 2). This induces T-cell apoptosis or anergy and allows tumor cells to evade the immune response [70,71]. Higher expression of PD-L1 is associated with a higher grade and worse outcome [70]. Immune checkpoint inhibitors that block the interaction of PD-1 with PD-L1 can potentially reinstate a T-cell antitumor immune response [71,72]. Here we review major trials studying PD-1 and PD-L1 inhibitors in glioblastoma (Table 3).

Nivolumab is a human immunoglobulin G4 monoclonal antibody that targets PD-1. After it was approved for several solid tumors, it was studied in several trials for patients with GBM. The phase 3 trial CheckMate 143 demonstrated no improvement in mOS for recurrent GBM patients treated with nivolumab vs. bevacizumab (9.8 vs. 10.0 months, HR 1.04, *p* = 0.76) [73]. Subsequently, two large phase 3 trials investigated the role of nivolumab in newly diagnosed GBM. In the CheckMate 548 trial, the addition of nivolumab to radiotherapy and temozolomide did not improve OS (mOS 28.9 vs. 32.1 months in nivolumab vs. placebo group, respectively, HR 1.10) or PFS (10.6 months vs. 10.3 months, HR 1.1) in newly diagnosed GBM patients with methylated/indeterminate MGMT promotor [4]. The companion phase 3 CheckMate 498 trial was designed to compare OS for new unmethylated MGMT glioblastoma patients treated with either nivolumab + RT vs. temozolomide + RT. Notably, this trial also did not meet its primary endpoint of prolonged survival (mOS = 13.4 months vs. 14.9 months, HR 1.31, *p* = 0.0037), and results favored the control arm [5]. The use of dual checkpoint blockade, PD-1, and CTLA-4, is currently undergoing investigation in the NRG Oncology cooperative group in a randomized phase 2/3 trial BN007 (NCT04396860), which has completed the phase 2 accrual. Intracerebral administration of CTLA-4 and PD-1 inhibitors ipilimumab and nivolumab, respectively, in addition to intravenous nivolumab, has also been shown to be safe in a phase 1 trial, with further results pending [74]. One can question from a mechanistic perspective, however, the need for direct intracranial administration.

Pembrolizumab, another anti-PD-1 antibody, has been extensively studied in glioblastoma patients but has also shown to be ineffective at improving mOS. In the KEYNOTE-028 study, results from the phase 1 trial in recurrent PD-L1 positive glioblastoma patients revealed a response rate of 8%. Interpretation of the results is limited due to the small sample (*n* = 26) as well as the lack of an active comparator [75]. Pembrolizumab was then studied in conjunction with bevacizumab, with the rationale being the latter drug’s possible role in promoting immunotherapy by inhibiting VEGF [76,77,78,79]. In this phase 2 trial, patients were randomized to receive either pembrolizumab + bevacizumab vs. pembrolizumab monotherapy to assess PFS at 6 months. Pembrolizumab failed to improve both PFS and OS [80].

Investigations focused on PD-L1 targeting in glioma have been more limited in scope. Atezolizumab, an anti-PD-L1 antibody, was studied in a phase 1 trial. The overall response was 6%, with some long-term survivors [81]. The N2M2/NOA-20 phase 1/2 trial includes an arm investigating atezolizumab in conjunction with asinercept (APG101), a CD95-Fc fusion protein intended to block apoptosis of activated T cells [82]. Preliminary results of another atezolizumab trial (NCT03174197) suggest that survival outcomes may be associated with differences in the microbiomes of patients [83]. These results have been presented only in abstract form thus far.

These results with checkpoint blockade are not terribly surprising given a number of considerations. First and foremost, glioblastoma patients are lymphopenic, with the T cells sequestered in the bone marrow [8]. Even upon gaining access to the glioblastoma tumor microenvironment (TME), the T cells are exhausted and are unlikely to be reinvigorated with immune checkpoint blockade or other types of immune suppression modulation [84,85]. This is coupled with the immunosuppressive “cold” tumor microenvironment as detailed in the following few sections, overall resulting in low immunogenicity. Additionally, PD-L1 is not frequently expressed in glioblastoma, and even those that do express PD-L1 do not necessarily fare better. In fact, PD-L1 expression was shown to be correlated with an increased risk of death [86]. Furthermore, there are many other redundant mechanisms of immune suppression that do not involve the PD-1/PD-L1 axis, rendering checkpoint inhibitors targeting this specific axis ineffective [85,86,87]. Lastly, limited drug availability due to the blood–brain barrier may impair the response of glioblastoma to checkpoint inhibitor therapy. 

Despite all of this, there are several studies that have demonstrated the potential benefit of neoadjuvant anti-PD1 immunotherapy in patients with glioblastoma. One study demonstrated that neoadjuvant pembrolizumab, in addition to adjuvant therapy following surgery, had prolonged OS compared to patients who received post-surgical therapy alone and also had increased T-cell expression and PD-L1 induction in the tumor microenvironment [88]. Another phase II trial demonstrated that neoadjuvant followed by adjuvant nivolumab led to immunomodulatory effects with improved immune cell infiltration and TCR diversity [89]. One should keep in mind that the studies demonstrating this benefit have all been relatively small in size. However, proof of principle exists. Neoadjuvant anti-PD1 immunotherapy has also shown promising results in other solid tumors, particularly in melanoma [90]. High rates of both radiographic and pathologic response with the neoadjuvant addition of nivolumab to anti-CTLA-4 inhibition (ipilimumab) in patients with melanoma have been demonstrated [91]. 

### 2.3. Targeting Regulatory T Cells and the Associated Immune Suppression Axis

The immunosuppressive tumor environment in GBM provides resistance to the anti-GBM immune response. Regulatory T cells (Tregs; CD4+ FoxP3+ CD25+) are regulatory T cells that promote and maintain the immunosuppressive tumor microenvironment [92] and can be activated by the immunosuppressive mediator, indoleamine 2,3 dioxygenase 1 (IDO1) [93]. GBM-cell IDO1 promotes tumorigenesis and increases immunosuppressive Treg recruitment while simultaneously decreasing cytotoxic CD8+ T-cell frequency, as shown in Figure 3 [94]. IDO1 expression is correlated with increasing tumor grade, as well as the expression of other immunosuppressive mediators such as PD-L1 [95]. Furthermore, glioma patients with upregulated intratumoral IDO1 expression have shorter OS when compared to those with intermediate and downregulated IDO expression (44.3 vs. 34 vs. 24.9 months) [95]. Interestingly, animal studies demonstrate that the genetic ablation of IDO1 by glioma cells results in decreased Treg recruitment [95] and significantly improved survival rates (150 days vs. 26.5 days in IDO1-competent GBM mice, *p* < 0.002) [95], pharmacological IDO1 enzyme inhibitors fail to recapitulate the same effect [96]. However, when IDO1 enzyme inhibition is used in combination with concurrent radiation and PD-1 mAbs, there is a synergistic improvement in the mOS [96]. This combination therapy also increased the CD8+ T cell: Treg ratio and provided long-lasting tumor control in 30–40% of mice with GBM [96]. Newer research has demonstrated that advanced age itself not only worsens the prognosis for GBM patients but can also suppress the efficacy of immunotherapies [96,97]. A phase 1 trial evaluating the combination of IDO1 inhibition with BMS-986205, PD-1 blockade with nivolumab, and concurrent radiotherapy with vs. without TMZ is currently underway [98].

IDO1 (indoleamine 2,3 dioxygenase 1) promotes the immunosuppressive tumor microenvironment through multiple mechanisms, including activation of MDSCs (myeloid-derived suppressor cells), Tregs (regulatory T cells), and M2 tumor-associated macrophages, while decreasing recruitment of NK (natural killer) cells and cytotoxic CD8+ T cells. 

Another Treg targeting therapy is agonistic antibody treatment against glucocorticoid-induced TNFR-related protein (GITR). Agonism of the GITR receptor has shown preclinical efficacy in a number of GBM models by either depleting or reprogramming Tregs [99,100,101]. This is currently being tested in the context of stereotactic radiotherapy and anti-PD1 administration in recurrent GBM (NCT04225039). Of note, Tregs are not a dominant mechanism of immune suppression in GBM, and many tumors lack any appreciable infiltration of this population [102].

### 2.4. T-Cell Activators

While checkpoint inhibitors and Treg targeted therapies are certainly the most widely examined immunotherapeutic, a newer class of T-cell activators is being explored in clinical studies of GBM. One phase 1/1b study uses an agonistic antibody against the T-cell co-activator 41BB (PF-05082566) in conjunction with CCR4 blockade for GBM and other solid tumors [103,104]. OX-40 ligand (OX-40L) is another T-cell activator being explored. In an ongoing phase 1 trial (NCT03714334), the oncolytic adenovirus DNX-2240, which encodes for OX-40L, will be administered to recurrent GBM patients. A third T-cell co-activator being targeted is the CD40/CD40L axis. An Fc-engineered CD40 agonist antibody (2141-V11) will be administered in combination with the EGFRvIII immunotoxin (D2C7-IT) in recurrent GBM patients in one phase I study (NCT04547777) [105]. All of these trials are still in their infancy, with results expected over the next few years. 

In summation, while targeting or enhancing T-cell activity is the largest avenue of immunotherapy clinical research in GBM, the inherent resistance of these tumors to immunotherapy may provide another hurdle to their successful use in patients. A summary of ongoing clinical trials investigating regulatory T cells and T-cell activators for glioblastoma is listed in Table 4.

## 3. Clinical Factors That Prevent Immunotherapeutic Efficacy in GBM

There are numerous GBM-intrinsic factors and SOC treatment-associated factors that influence the efficacy of immunotherapy. Temozolomide (TMZ), an SOC alkylating chemotherapy, induces leukopenia, and specifically, lymphopenia [106]. The risk increases with steroid usage and concomitant use of radiation [1]. Treatment-associated lymphopenia can persist for up to one year and subsequently dampen the effect of subsequent immunotherapies [107]. Furthermore, the dose of alkylating therapy or route of administration may play a critical role in immunotherapeutic outcomes [108,109].

Surgical management also impacts the response to immunotherapy. The reduced number of glioblastoma tumor cells that remain after gross total resection may result in fewer tumor antigens and, consequently, a less robust immune response. Furthermore, SOC treatments also have the potential to select aggressive tumor cell lines after targeting the less resistant cells, which can negatively impact the response to immunotherapies. Additionally, dexamethasone, a commonly used drug for treating symptomatic vasogenic edema, impairs proinflammatory CD8+ T-cell responses in GBM patients, thus contributing to a weakened immune response. Treatments such as surgery or radiation have also been shown to activate a host of immunosuppressive feedback loops [110] that may lead to a more pronounced degree of immunosuppression than can be observed prior to therapeutic intervention [1].

Intratumoral heterogeneity remains one of the most difficult impediments for glioblastoma therapies to overcome. Using single-cell RNA sequencing, it has been demonstrated that a single primary glioblastoma consists of a heterogeneous milieu of cells from a variety of different subgroups [111]. This intratumoral variability exists on both a temporal and spatial level, the latter of which highlights the risk of sampling error with a single regional biopsy [112] and has implications for both biomarker identification as well as therapeutic response. While treatment may be highly successful in eradicating one subclone, it may be unable to address the various other cellular subpopulations and, furthermore, may select subpopulations that become resistant to subsequent therapy [113,114,115,116]. Intratumoral heterogeneity highlights the potential need for multimodal, combinatorial therapies for glioblastoma. 

In contrast, some aspects of SOC treatment may be beneficial with respect to immunotherapeutic approaches. For example, radiotherapy promotes effector T-cell mobilization into tumors and can help suppress the proliferation of immunosuppressive regulatory T cells. Furthermore, tumor cell death from radiotherapy can increase the exposure to a larger amount of immunostimulatory antigens to promote an immune response [1,117,118,119]. 

Of note, due to the location and nature of the disease, it is not feasible to obtain tumor tissue samples from all timepoints of interest. In turn, our understanding of the temporal immunologic picture of the tumor and its microenvironment is incomplete [120].

## 4. Predictive Biomarkers of Immunotherapeutic Efficacy

Numerous potential predictive biomarkers for response to immunotherapies in GBM have been investigated. These include: PD-L1 expression, tumor mutational burden (TMB), PTEN and MAPK pathway mutations, as well as the replication stress response (RSR). These are summarized in Table 5. A limitation to these biomarker analyses, alluded to above, is a lack of understanding of the optimal timepoints in the disease course for these to be assessed for validation. It is likely that different biomarkers, if valid, will have distinct predictability at specific disease course timepoints.

### 4.1. PD-L1 Expression

PD-L1 expression has been shown to impact the response to immunotherapies in several solid cancers such as melanoma and non-small-cell lung cancer, where PD-L1 expression on these tumor cells (i.e., presence of target) was shown to be associated with objective treatment response to PD-1 inhibition [121,122]. Expression of PD-L1 in GBM is heterogeneous, and higher expression is correlated with significantly shorter survival [91]. However, the expression of PD-L1 does not appear to be strongly predictive of response to immunotherapy in GBM patients. In addition, there is a lack of association between PD-1/PD-L1 expression and the tumor mutational burden of glioblastoma tumors [123].

### 4.2. Tumor Mutational Burden

Tumor mutational burden (TMB) refers to the total number of mutations within the genome of a tumor cell [123]. Increased TMB can be associated with increased prevalence of immunogenic epitopes of tumor cell surfaces, which, in theory, could improve response to immunotherapy [123,124]. However, not all mutations, including those for genes encoding intracellular proteins, are immunogenic, and the TMB may be heterogeneous between tumor cells. Finally, there may be aspects intrinsic to the tumor microenvironment, which may supersede the importance of the TMB and immunogenic epitope expression. 

Increased TMB has been previously shown to be associated with an improved response to PD-1 blockade in several cancer types [124]. However, recent studies demonstrated that high TMB in glioblastoma is not broadly predictive of response to immune checkpoint blockade and that samples taken from tumors responsive to anti-PD1 therapy do not have a higher rate of somatic mutations [125,126]. In one study, hypermutated glioma patients treated with PD-1 blockade had similar PFS and OS times compared to patients with non-hypermutated gliomas [127]. Additionally, patients with hypermutated gliomas treated with PD-1 blockade have shorter mOS compared to historical controls treated with other systemic agents [127]. Taken together, these results are suggestive that, in contrast to other cancers, GBM TMB plays a minimal role in response to immunotherapy and, in fact, may negatively impact the response. It appears that an elevated neoantigen load may need to be accompanied by increased immune infiltration for this type of approach to be effective [128]. Of note, there are case reports of germline mutations in DNA repair enzymes in brain tumor patients that were associated with favorable responses to immune checkpoint inhibition, presumably due to an increased TMB [129,130]. However, the elevation of TMB from germline mutations does not necessarily equate to a more homogeneous presence of specific mutations. These results are not generalizable to all glioblastoma patients, as the overwhelming majority of patients with brain tumors due not harbor germline DNA repair enzyme mutations.

### 4.3. PTEN Mutations

PTEN, a negative regulator of the PI3K/Akt pathway, has long been known to be a key contributor to the immunosuppressive microenvironment of GBM [131]. PTEN mutations in GBM also appear to be associated with a reduced response to anti-PD-1 therapy. PTEN mutations have been shown to occur more frequently in non-responders to PD-1 blockade as opposed to responders [126]. Furthermore, PTEN mutations may help to promote an immunosuppressive microenvironment by increasing tumor cell clustering that impedes immune cell infiltration, as well as increased expression of CD44, a cell surface adhesion receptor that facilitates cell interactions with the extracellular matrix. This results in increased tumor cell migration [126]. This corroborates findings from prior melanoma studies demonstrating that a loss of PTEN expression results in increased immunosuppressive cytokine expression, decreased T-cell infiltration, and subsequent decreased T cell-mediated cell death [132].

It should be noted that a large majority of glioblastomas exhibit chromosome 10q deletion, which is where the PTEN gene is located [133,134]. Thus, loss of PTEN expression and function as a result of chromosome 10q deletion, as opposed to PTEN mutation, likely contributes to the lack of therapeutic response in many glioblastomas.

### 4.4. MAPK Pathway Aberrancies

On the other hand, MAPK pathway mutations (including BRAF and PTPN11) in GBM are associated with an improved response to PD-1 blockade, defined in one study as either stable or shrinking GBM volume on imaging or a minimal amount of tumor cells in tissue samples [126]. In this study, GBM patients who were responsive to anti-PD-1 therapy had a significantly higher rate of BRAF and PTPN11 mutations compared to non-responders [126]. These mutations are known to drive MAPK/ERK pathway signaling, which results in an end product of ERK1/2 phosphorylation (*p*-ERK) [135]. Though BRAF/PTPN11 mutations were highly enriched in the responder patients to PD-1 blockade, only 30% of these patients had such mutations. Recently, we showed that phosphorylation of ERK1/2 (pERK), a downstream effector of MAPK, is present and elevated in all responders to PD1 blockade and in those that had prolonged survival following this form of immunotherapy [136]. Yet, not all patients with elevated pERK had long-term survival following this form of immunotherapy. The correlation between pERK with survival following PD-1 blockade was observed in two independent patient cohorts [136]. Responder patients that had elevated pERK included those without BRAF/PTPN11 mutations. In addition, increased *p*-ERK resulted in a higher number of myeloid cells and microglia expressing MHC-II in the tumor microenvironment, suggesting a proinflammatory effect [136]. It is uncertain if this may apply to other immunotherapeutic approaches. Interestingly, preclinical glioma models have shown that gliomagenesis in the absence of CD8 T cells leads to an increase in pERK in the tumors [137]. This may suggest that tumor-targeting immune infiltrates may facilitate the over-activity of immunosuppressive negative feedback loops.

### 4.5. Replication Stress Response

A new focus of the investigation has been the replication stress response (RSR), which is used by cells to slow DNA replication when replication stress, such as DNA lesions or limited nucleotides, occurs and requires repair [138]. Utilizing murine breast cancer models, one study demonstrated that gene expression of RSR defects (RSRDs) not only led to the accumulation of immunostimulatory cytosolic single-stranded DNA and promoted dendritic and T-cell infiltration but also was predictive of immune checkpoint blockade (ICB) response. The RSRD score was shown to be generalizable across tumor subtypes and was found to be a predictive and specific biomarker for clinical response to ICB in GBM patients [139].

**Table 5 cancers-14-04023-t005:** Studies investigating factors affecting checkpoint inhibitor response in glioblastoma.

Predicted Response to Checkpoint Blockade	Target	Authors	Primary Results	PMID
No effect	PD-L1 expression	Hodges et al.	There is no association between tumor mutational load and PD-1/PD-L1 expression (*p* = 0.7699, *p* = 0.8237)	28371827 [123]
No effect/negative response	Tumor mutational burden	McGrail et al.	Gliomas with high TMB had a low ORR (15.3%, 95% CI 92–23.4). ICB-treated glioma patients had worse OS vs. those treated with other modalities (*p* = 0.23 × 10^−5^)	33736924 [125]
		Touat et al.	Patients with hypermutated gliomas treated with PD-1 blockade had similar PFS and OS vs. those with non-hypermutated gliomas (1.38 vs. 1.87 mo, 8.7 vs. 9.96 mo). Patients with hypermutated gliomas had shorter mOS with PD-1 blockade vs. other treatments (8.07 vs. 16.10 mo, *p* = 0.02)	32322066 [127]
Negative response	PTEN mutations	Zhao et al.	PTEN mutations occur more frequently in non-responders to PD-1 blockade vs. responders (*p* = 0.0063, OR = 8.5)	30742119 [126]
Improved response	MAPK pathway mutations	Zhao et al.	MAPK pathway alterations (PTPN11, BRAF) are enriched in responders to PD-1 inhibitors (Fisher *p* = 0.019, OR = 12.8)	30742119 [126]
		Arrieta et al.	ERK1/2 activation in recurrent GBM predicts OS after PD1 blockade (HR = 0.18, 95% CI 0.06–0.56)	35121903 [120]
	Replication stress response defects (RSRD)	McGrail et al.	RSR defects lead to immunostimulatory cytosolic ssDNA and improved ICB response (*p* = 0.00019 in breast cancer model, *p* < 0.1 × 10^−6^ in GBM cohort)	34705519 [139]

MAPK = mitogen-activated protein kinase; RSRD = replication stress response defects; PD-1 = programmed death 1; PD-L1 = programmed death ligand 1; TMB = tumor mutational burden; GBM = glioblastoma; PTEN = phosphatase and tensin homolog; OS = overall survival; PFS = progression-free survival; HR = hazard ratio; OR = odds ratio; ORR = overall response rate; ICB = immune checkpoint blockade.

## 5. Making Things Hot: Promoting T-Cell Infiltration into GBM

Glioblastomas are immunologically “cold” tumors that promote an overall immunosuppressive tumor microenvironment with few lymphocyte infiltrates and impaired antigen presentation [140]. These changes help explain why glioblastomas respond poorly to checkpoint inhibitors. In addition, when infiltrates occur, as suggested above, they may overactivate innate immunosuppressive systems at times. Several studies are now aimed at converting “cold” glioblastomas into ones that are “hot.” We have summarized some of these studies in Table 6.

### 5.1. Oncolytic Viral Therapies

Oncolytic viral therapy uses viruses to selectively target tumor cells and subsequently induce tumor lysis, promote antigen presentation, recruit tumor-infiltrating lymphocytes, and activate the innate immune response [141,142,143]. A total of over 15 DNA and RNA virus species are currently being studied in glioma, including adenovirus, parvovirus, enterovirus, and herpes simplex virus-1 (HSV-1), among others [144]. Recently, after being studied in a phase II trial, one genetically engineered oncolytic virus based on HSV-1 (G47D) (Delytact/Teserpaturev) was conditionally approved for the treatment of malignant glioma in Japan. Additional details on current research involving oncolytic viral therapy are well summarized separately [144,145].

### 5.2. STING Agonism

STING (stimulator of interferon genes) agonists, through IFN-b and cytokine production, promote T-cell infiltration into the tumor microenvironment and thus have been of key interest in recent studies aimed at improving immunotherapeutic responses to checkpoint inhibitors [146,147,148,149]. A recent study demonstrated that intratumoral administration of a STING agonist (IACS-8779) in canines with glioblastoma was feasible, safe, and resulted in a dose-dependent radiographic response [150]. These promising results warrant further investigation in additional translational studies. 

### 5.3. STAT3 Signaling

Signal transducers and activators of transcription 3 (STAT3) is an intracellular cell signaling molecule that is highly expressed and constitutively activated in a variety of malignancies, including GBM [151]. STAT3 activation promotes the production of factors such as IL-10, IL-23, and TGF-b, which results in the accumulation of immunosuppressive cells such as regulatory T (Treg) cells, M2 tumor-associated macrophages (TAMs), and myeloid-derived suppressor cells (MDSC), as well as the inhibition of dendritic cell development, ultimately promoting the immunosuppressive tumor microenvironment [151,152] as shown in Figure 4. As a result, STAT3 has become an emerging target for immunotherapy. 

Several drugs targeting STAT3 activation, either through upstream kinase inhibition via JAK1/2 targeting (sorafenib, AG490, SAR317461) or through blocking of STAT3 phosphorylation (oleanolic acid, embelin), have been developed. Preclinical studies have demonstrated that these drugs decrease the proliferation, migration, and angiogenesis of glioblastoma cells and promote cytokine release for an improved immune response [153,154,155]. One study demonstrated that the STAT3 inhibitor WP1066 induced immunostimulatory cytokine release to promote T-cell effector function in murine models while also decreasing tumor-mediated immunosuppressive mechanisms of Treg and M2 macrophage accumulation [155]. WP1066 is now being evaluated in clinical trials, including for pediatric patients (NCT04334863), and has been shown to be well tolerated in human subjects in doses of 8 mg/kg, which exceeds the amount needed for conducting phase II studies in combination with radiation therapy [156]. The combination of radiation and STAT3 blockade with WP1066 in preclinical models markedly influenced the TME by inducing DC and T-cell infiltration within the tumor [157]. Ultimately, these results highlight how STAT3 antagonism can alter the immunosuppressive tumor microenvironment and subsequently improve the antitumor immune response.

STAT3 (signal transducers and activators of transcription 3) promotes the immunosuppressive tumor microenvironment through multiple mechanisms, including increased secretion of IL-10 and IL-23 (interleukin-10 and interlekin-23), MDSCs (myeloid-derived suppressor cells), and M2 tumor-associated macrophages, as well as inhibiting the development of dendritic cells. Recently studied methods of targeting STAT3 activation include inhibiting JAK1/2, as well as blocking the STAT3 phosphorylation. 

### 5.4. Cytokine Therapies

Interleukin-12 (IL-12) has also been an important immune-activating cytokine under investigation as a form of immunotherapy for glioblastomas. Preclinical and clinical studies have shown that IL-12-based gene therapy can lead to safe and effective expression of this potent cytokine in gliomas and in the peri-tumoral brain [158,159,160,161]. IL-12 is known to help mediate both natural and adaptive immunity, increasing the production of interferon-y (IFN-y) as well as CD8+ T-cell production [161,162,163,164,165,166]. Previous studies investigating IL-12 therapy were limited due to toxicity and tolerability. However, recent studies have turned toward using a viral vector gene therapy with a ligand-inducible expression switch to help control IL-12 production in the local environment. Veledimex (VDX) is an activator ligand designed to promote the production of IL-12. A phase 1 trial demonstrated that VDX administration resulted in a dose-dependent increase in both IL-12 and IFN-y serum concentrations in patients with high-grade gliomas. Additionally, tissue analysis after treatment demonstrated a 17-fold increase in CD3+ T cells, as well as increases in CD8+ T cells, PD-1 expressing immune cells, and PD-L1 expressing cells. Preliminary analyses demonstrated a positive association between VDX dose and OS (*p* = 0.0004). These results indicate that IL-12 therapy helps transform an immunologically “cold” tumor microenvironment into a proinflammatory “hot” environment with an increased influx of immune cells [159]. IL-12 is incorporated in another multifaceted approach using synthetic DNA plasmids currently undergoing investigation in newly diagnosed glioblastoma (NCT03491683) [167].

### 5.5. Disrupting the Blood–Brain Barrier to Enhance Immunotherapy

Drug delivery into the CNS is notoriously difficult due to the protective blood–brain barrier (BBB), which provides another explanation for why so many investigated therapies for recurrent GBM have failed.

BBB disruption can occur with radiation therapy. Prior studies have demonstrated that low dose radiation up to 20–30 Gray can alter the integrity of the BBB and improve the permeability to certain therapeutic drugs [168]. One group demonstrated that CSF concentrations of methotrexate increased three-fold after irradiation therapy [169]. However, studies have now shifted more toward using ultrasound as a method of disrupting the BBB. 

The combination of pulsed ultrasound and intravenous microbubble injection has been shown to be a safe, minimally invasive, well-tolerated, and reversible method of disrupting the BBB [170]. Two major strategies have been implemented to deliver US-based BBB disruption: an extracranial device that delivers focused ultrasound called the ExAblate system and an implantable ultrasound device called the SonoCloud. Both have been studied in multiple preclinical and clinical trials [170,171,172,173]. A newer version of the SonoCloud, called the SonoCloud-9, is comprised of nine transducers on an implantable grid and is designed to cover a larger resection area. It is currently being studied in a multicenter phase 1/2a clinical trial [174]. 

Several studies have investigated the impact of US-mediated BBB disruption on drug delivery to the CNS. Due to its poor CNS penetration despite robust in vitro anti-glioma effects, the microtubule-stabilizing agent paclitaxel was recently of particular interest. The combination of US-based BBB disruption followed by an albumin-bound formulation of paclitaxel (Abraxane) not only led to a significant 3–5-fold increase in paclitaxel concentration at multiple time points compared to non-sonicated controls but also improved mOS compared to Abraxane only (35 vs. 31 days, 0.0036) [175]. US-based BBB disruption, therefore, can both enhance CNS penetration of PTX and improve survival [176]. Additionally, paclitaxel has been recently identified as an inducer of immunogenic cell death, a form of cell death that directly promotes antitumor immunity [176], and as a result, BBB-mediated delivery of paclitaxel may enhance immunotherapies for GBM [177].

Animal model studies have examined the impact of low-intensity pulsed ultrasound (LIPU) on various drug efficacies. In mice, the addition of LIPU to anti-PD 1 therapy resulted in a longer mOS compared to anti-PD 1 therapy alone (58 vs. 39 days, respectively), though it did not reach significance [178]. Mice treated with both LIPU and CAR T-cell therapy had significantly increased CNS CAR T-cell delivery and longer mOS compared to CAR T-cell therapy alone. Furthermore, the addition of LIPU to an APC-based therapy demonstrated a significantly increased deposition of CXCL10-secreting APCs in the tumor microenvironment, as well as enhanced survival (*p* < 0.05) [178]. 

## 6. Harnessing the Innate Immune System for GBM Immunotherapy

The innate immune system, which constitutes our first line of defense, mediates broad responses against pathogens while also activating adaptive immunity for more specific targeting. Research has now shifted additional attention to methods of modulating the innate immune system for the treatment of glioblastoma. Recent glioblastoma clinical trials focused on the innate immune system are listed in Table 7.

### 6.1. Tumor-Infiltrating Myeloid Cells

Tumor-infiltrating myeloid cells are the most abundant cellular infiltrates in GBM [179,180], which can sometimes comprise up to 50% of the tumor mass [179]. These heterogeneous groups of cells are interchangeably given names such as: myeloid-derived suppressor cells (MDSC), monocyte-derived macrophages (MDM), tumor-associated macrophages (TAM), and tumor-associated myeloid cells (TAMC). They have become an incredibly attractive target for GBM immunotherapy due to their central role in promoting immunosuppression in GBM [181,182,183]. MDSCs can be subdivided into monocytic stem cells (mMDSC) versus granulocytic stem cells (gMDSC), which have distinct sex-dependent roles in GBM pathogenesis. Recent research has shown that mMDSCs accumulate intratumorally and promote GBM progression in males, whereas gMDSCs accumulate peripherally and primarily regulate immune suppression in females [184]. While large trials of immunotherapies for glioblastoma have thus far not demonstrated any clear sex-based differences in outcomes, it is possible that this may still prove to be of importance as we explore immunotherapeutic management of glioblastoma.

There are two main methods of myeloid targeting in GBM. The first involves depleting or preventing the recruitment of cells to the tumor site, and the second involves “reprogramming” these cells to become immunostimulatory (Figure 5). 

Approaches that deplete myeloid cells from GBM have shown potential in preclinical studies of GBM. One study showed that nanoparticle-mediated targeting of myeloid cells using PD-L1 antibodies robustly depletes myeloid cells after radiation, leading to pronounced survival benefits [115]. Other studies have utilized blockade of CCR2 as a modality to prevent myeloid recruitment to the CNS [185]. Recent work has also identified that the metabolic inhibitor difluormethylornithine (DFMO) could be used to deplete myeloid cells from GBM and promote an inflammatory tumor microenvironment [186]. Clinically, a recent phase 0/1 trial utilized the natural sensitivity of MDSC to 5-Fluorouracil to deplete this population from the tumors of GBM patients [187]. 

Reprogramming myeloid cells to a more tumor supportive phenotype has gained attention by targeting the colony-stimulating factor 1 receptor (CSF1R). Pioneering preclinical work identified that pharmacological blockade of CSF1R using BLZ925 can promote the survival of animals with GBM [188], which can be used in combination with stereotactic radiotherapy for even more robust antitumor responses [189]. This led to a phase 1 exploration of BLZ945 in combination with PD-1 blockade for solid tumors (including GBM) [190]. Another CSF1R inhibitor, PLX3397 (pexidartinib), was tested in recurrent GBM and shown to be safe, though it did not result in any changes in OS [191]. Strategies to target this compartment in GBM are of continued interest to researchers but will likely need to be a component of a multi-agent regimen.

The presence of immunosuppressive macrophages has been a difficult hurdle to overcome in the attempts to develop effective immunotherapies for glioblastoma. Some recent work has focused on using niacin (vitamin B3) to stimulate the antitumor activity of macrophages and microglia. One study demonstrated that the exposure of monocytes to niacin resulted in increased levels of interferon-a14 for a subsequent anti-proliferative effect. Furthermore, preclinical work with niacin-treated mice with brain tumor-initiating cells led to increased monocyte and macrophage intratumoral infiltration, reduced tumor size, and improved survival [192]. A phase I/II study studying the efficacy of niacin in addition to radiation and temozolomide is currently ongoing (NCT04677049). 

### 6.2. NK Cells

Natural killer (NK) cells are characterized by the expression of CD16 and CD56 surface antigens and lack CD3/T-cell receptor molecules. In contrast to T cells, NK cells do not require antigen sensitization prior to killing targets [193]. Although NK cells can migrate into the GBM microenvironment, these tumor-infiltrating NK (TI-NK) cells are significantly altered in such a fashion that their cytotoxic function is impaired, allowing glioblastomas to evade NK cell targeting. NK cell dysfunction occurs through direct contact between glioblastoma stem cells and NK cells, which results in the av integrin-mediated release of TGF-β1. Preclinical studies have shown that inhibition of the av integrin/TGF-β1 axis can protect NK cell antitumor activity, and co-administration of NK cells with either TGF-β or av inhibitors resulted in improved OS in mice [194]. These results are overall suggestive of a promising new immunotherapy strategy for GBM patients.

### 6.3. Gamma Delta T Cells

Gamma delta (γδ) T cells play a role in both the innate and adaptive systems and are characterized by T-cell receptors comprised of one gamma and one delta chain, in contrast to other T cells with alpha and beta chains. Though they are primarily found in the gut mucosa, their role in impairing tumorigenesis has been an evolving area of interest [195]. Preclinical animal studies had previously demonstrated that γδ T cells have a cytotoxic effect on GBM cells [196,197,198]. When compared to control patients, one study demonstrated that GBM patients had a lower absolute γδ T-cell count and decreased γδ T-cell proliferation. Though tumor infiltration by γδ T cells may be hindered by the lymphodepletive effects of temozolomide, new studies have focused on modifying γδ T cells to be resistant to temozolomide. Preliminary results of a phase 1 trial investigating the combination of modified γδ T cells and temozolomide in new glioblastoma patients, known as drug-resistant immunotherapy, showed the combination therapy to be safe and feasible [199,200]. Further results are pending.

### 6.4. The Role of B Cells

Although most strategies for cancer treatments are focused on the adaptive immune system and specifically T cells, recent work has highlighted the potential role of B cells, which are involved in both the adaptive and innate immune systems. Intratumoral B cell infiltration is associated with a suitable prognosis in several solid tumors, such as melanoma [201,202], breast cancer [203], colorectal cancer [204], and non-small cell lung cancer (NSCLC) [205], and many others [206,207,208]. B cells that express the co-stimulatory marker 4-1BBL have been studied in GBM models. Signaling via 4-1BBL is associated with decreased T-cell death, enhanced T-cell proliferation, and improved immunological memory. 4-1BBL is expressed on CD4+ and CD8+ T cells as well as B cells. B cells that express 4-1BBL have been shown to enhance cytotoxic CD8+ T-cell response [209]. In one recent study, 4-1BBL+ B cells that were activated via CD40 and IFNγ stimulation (known as the BVax vaccine) were shown to have increased MHC-I and MHC-II surface expression for both CD8+ T-cell activation and antigen presentation as an APC. Furthermore, the combination of BVax + RT + TMZ + anti-PD-L1 therapy improved overall survival, resulted in tumor eradication in 80% of mice, and inhibited tumor recurrence after reinjection. In early translational studies, BVax derived from GBM patients’ activated autologous CD8+ T cells have successfully killed autologous glioma cells in an antigen-specific manner. Overall, these results support further investigation in future translational studies [210].

## 7. Finding the Balance

The interplay between the immune system and glioblastoma is complex and influenced by a myriad of factors. Certain molecular factors that predict a favorable response to immunotherapies in non-CNS tumors paradoxically portend a worse response in glioblastoma. Thoughtful consideration of these factors can help guide our choice of various synergistic combination therapies that take advantage of the vast immune landscape in a multipronged approach. Therapies that both improve immunostimulatory responses against glioblastoma while decreasing the immunosuppressive response will likely be the most successful. 

## 8. Conclusions

The prognosis for glioblastoma patients has unfortunately remained poor with our current SOC therapeutic options. Though immunotherapy has proved successful in treating other cancers, results in clinical trials for glioblastoma patients have been rather disappointing. Molecular factors influencing the response to immunotherapy have been extensively studied and should be incorporated in future translational and clinical studies. Additionally, other strategies for enhancing responses to immunotherapies, including promoting an immunostimulatory tumor environment and utilizing the innate immune system, should be areas of an increased focus to develop more effective immunotherapies for glioblastoma patients.

## Figures and Tables

**Figure 1 cancers-14-04023-f001:**
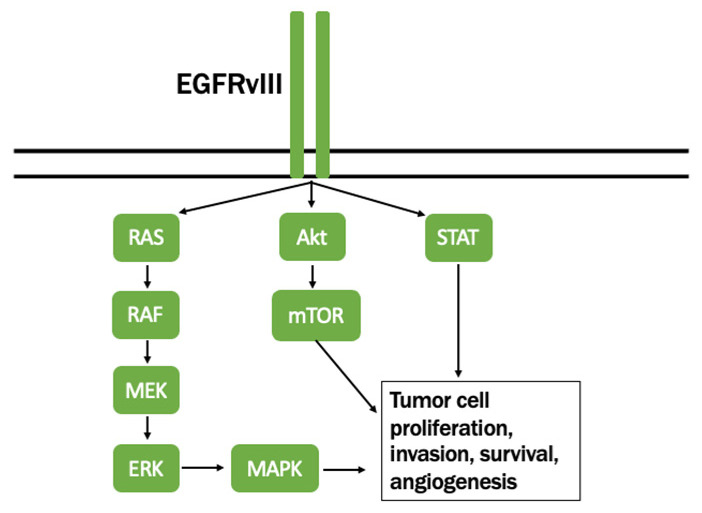
EGFRvIII signaling pathways.

**Figure 2 cancers-14-04023-f002:**
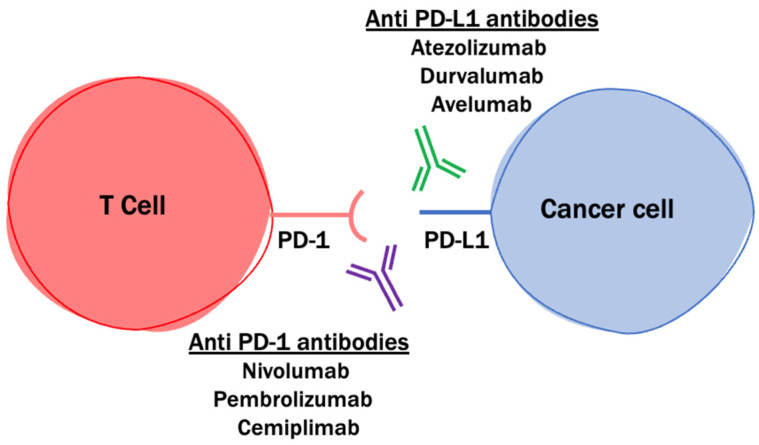
PD-1 and PD-L1 expression. PD-1 (programmed death-1) and PD-L1 (programmed death ligand 1) are expressed in T cell and cancer cells, respectively. Examples of PD-1 blockers include nivolumab, pembrolizumab, and cemiplimab. Examples of PD-L1 blockers include atezolizumab, durvalumab, and avelumab.

**Figure 3 cancers-14-04023-f003:**
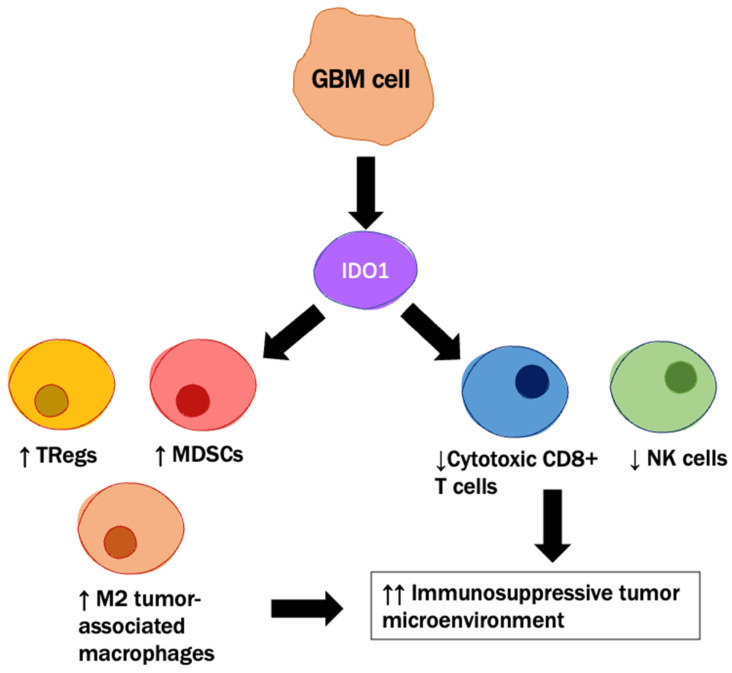
IDO1 promotes the immunosuppressive tumor microenvironment through multiple mechanisms. ↑, ↑↑ = increased, ↓ = decreased.

**Figure 4 cancers-14-04023-f004:**
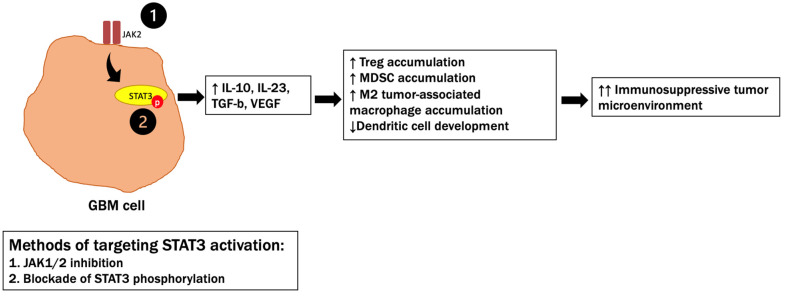
STAT3 promotes the immunosuppressive tumor microenvironment through multiple mechanisms. ↑, ↑↑ = increased, ↓ = decreased.

**Figure 5 cancers-14-04023-f005:**
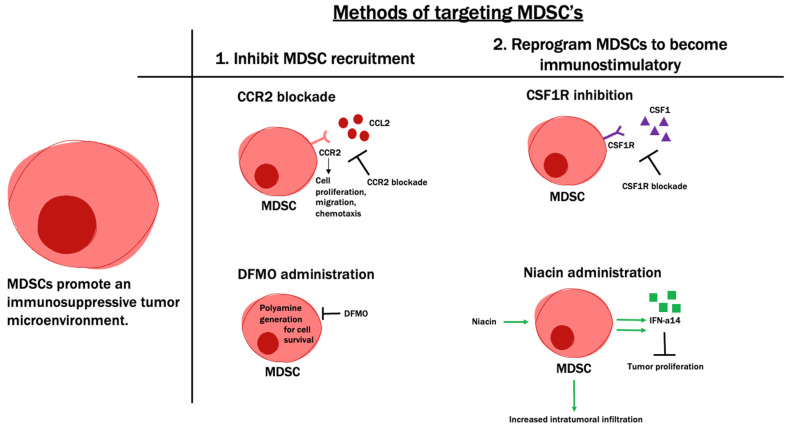
MDSCs can be targeted through various mechanisms. MDSCs (myeloid-derived suppressor cells) promote an immunosuppressive tumor microenvironment. Two main methods of targeting MDSCs include (1) inhibiting MDSC recruitment, such as with CCR2 (C-C chemokine receptor 2) blockade, as well as DFMO (difluormethylornithine) administration, and (2) reprogramming MDSCs to become immunostimulatory, such as with CSF1R (colony-stimulating factor 1 receptor) inhibition and niacin administration.

**Table 1 cancers-14-04023-t001:** Completed and ongoing peptide vaccine clinical trials for GBM.

**Completed Trials**						
**Vaccine Type/Target**	**Study Title**	**Phase**	**Size**	**Intervention Details**	**Primary Endpoint**	**Reference**
EGFRvIII	Phase III Study of Rindopepimut/GM-CSF in Patients With Newly Diagnosed Glioblastoma (**ACT IV**)	3	745	Rindopepimut + TMZ vs. TMZ	OS = 20.1 (95% CI 18.5–22.1) vs. 20.0 mo (95% CI 18.1–21.9)	NCT01480479
	A Study of Rindopepimut/GM-CSF in Patients with Relapsed EGFRvIII Positive Glioblastoma (**ReACT**)	2	73	Rindopepimut + bevacizumab vs. Bevacizumab	PFS at 6 mo = 28% vs. 16% (HR = 0.72, 95% CI 0.42–1.21)	NCT01498328
WT1	Phase II clinical trial of Wilms tumor 1 peptide vaccination for patients with recurrent glioblastoma multiforme	2	21	WT1 peptide	mPFS = 20 weeks 6 mo PFS rate = 33.3%	N/A
	A phase I study of the WT2725 dosing emulsion in patients with advanced malignancies	1	62	WT1 peptide (WT2725)	Maximum tolerated dose. * Response rate in GBM = 20% * OS12 in GBM = 33%	NCT01621542
	IMT-03 Clinical Trial for Newly Diagnosed Malignant Glioma with WT1-W10 Vaccination	1/2	27	WT peptide (W10)	PFS = 12.7 mo OS = 21.9 mo	N/A
IDH1	Targeting IDH1R132H in WHO Grade III-IV IDH1R132H-mutated Gliomas by a Peptide Vaccine—a Phase I Safety, Tolerability and Immunogenicity Multicenter Trial (NOA-16)	1	33	IDH1 peptide vaccine	Safety and immunogenicity. 93.3% with vaccine-induced immune response	NCT02454634
CMV	Vaccine Therapy in Treating Patients with Newly Diagnosed Glioblastoma Multiforme (**ATTAC**)	1/2	12	pp65-DC vaccine + Td preconditioning	Safety and feasability * OS = 20.6–47.3 mo vs. 13.8–41.3 mo (*p* = 0.013)	NCT00639639
	Long-term Survival in Glioblastoma with Cytomegalovirus pp65-Targeted Vaccination (**ATTC-GM**)	1	11	pp65 DC vaccine + GM-CSF	Safety and feasibility * mPFS = 25.3 mo vs. 8 mo (*p* = 0.0001), * mOS = 41.1 vs. 19.2 mo (*p* = 0.0001)	NCT00639639
	Evaluation of Overcoming Limited Migration and Enhancing Cytomegalovirus-specific Dendritic Cell Vaccines with Adjuvant TEtanus Pre-conditioning in Patients With Newly-diagnosed Glioblastoma (**ELEVATE**)	2	43	pp65 DC vaccine + TMZ vs. Pp65 DC vaccine + TMZ + preconditioning	3-year OS = 34% (95% CI 19–63%) vs. 6% (95% CI 1–42%)	NCT02366728
Multipeptide Vaccines	Efficacy finding cohort of a cancer peptide vaccine, TAS0313, in treating recurrent glioblastoma	1/2	9	TAS0313	Safety and efficacy. ORR = 11.1% (95% CI = 0.3–48.2%)	JapicCTI-183824
	A Randomized, Double-blind, Controlled Phase IIb Study of the Safety and Efficacy of ICT-107 in Newly Diagnosed Patients with Glioblastoma Multiforme (GBM) Following Resection and Chemoradiation	2	124	ICT-107 vs. placebo	OS = 17 vs. 15 mo (HR = 0.87, *p* = 0.58). * PFS = 11.2 vs. 9.0 mo (HR = 0.57, *p* = 0.011)	NCT01280552
Neoantigens	A Phase I Study of a Personalized NeoAntigen Cancer Vaccine With Radiotherapy Plus Pembrolizumab/MK-3475 Among Newly Diagnosed Glioblastoma Patients	1/1b	8	NeoVax + RT vs. Neovax + RT + Pembrolizumab	Safety and tolerability	NCT02287428
**Ongoing Trials**						
**Vaccine Type/Target**	**Study Title**	**Phase**	**Size**	**Intervention details**	**Primary endpoint**	**Reference**
Survivin	A Phase II Study of the Safety and Efficacy of SVN53-67/M57-KLH (SurVaxM) in Survivin-Positive Newly Diagnosed Glioblastoma	2	66	SurVaxM + TMZ	PFS at 6 mo	NCT02455557
	Phase II Study of Pembrolizumab Plus SurVaxM for Glioblastoma at First Recurrence	2	40	SurVaxM + pembrolizumab	PFS at 6 mo	NCT04013672
TERT	Anticancer Therapeutic Vaccination Using Telomerase-derived Universal Cancer Peptides in Glioblastoma (UCPVax-Glio)	1/2	56	UCPVax vs. UCPVax + TMZ	Anti-TERT T-cell response	NCT04280848
Multipeptide Vaccines	First-in-Human, Phase 1b/2a Trial of a Multipeptide Therapeutic Vaccine in Patients with Progressive Glioblastoma (ROSALIE)	1/2	52	EO2041 + nivolumab vs. EO2041 + nivolumab + bevacizumab	Safety and tolerability	NCT04116658

EGFRvIII = epidermal growth factor receptor variant III; GM-CSF = granulocyte-macrophage colony-stimulating factor; TMZ = temozolomide; RT = radiotherapy; WT1 = Wilms tumor 1; IDH1 = isocitrate dehydrogenase 1; CMV = cytomegalovirus; Td = tetanus toxoid; DC = dendritic cell; TERT = telomerase reverse transcriptase; OS = overall survival; mOS = median overall survival; PFS = progression-free survival; mPFS = median progression-free survival; HR = hazard ratio; CI = confidence interval; ORR = overall response rate. * Secondary endpoints.

**Table 2 cancers-14-04023-t002:** Autologous vaccine clinical trials for GBM.

Autologous Vaccine	Study Title	Phase	Size	Intervention	Primary Endpoint	Reference
Dendritic cells	A Phase III Clinical Trial Evaluating DCVax^®^-L, Autologous Dendritic Cells Pulsed With Tumor Lysate Antigen For The Treatment Of Glioblastoma Multiforme (GBM)	3	331	TMZ + DCVax-L vs. TMZ + placebo	mOS = 23.1 mo (95% CI 21.2–25.4)	NCT00045968
	Phase 1 Study of a Dendritic Cell Vaccine for Patients with Either Newly Diagnosed or Recurrent Glioblastoma	1	36	DC vaccine + GBM stem-like cell lysate	Safety and tolerability * mPFS = 8.75 mo in new GBM, 3.23 mo in recurrent GBM * mOS = 20.36 mo in new GBM, 11.97 mo in recurrent GBM	NCT02010606
HSP	Phase I/II Trial of Heat Shock Protein Peptide Complex-96 (HSPPC-96) Vaccine for Patients With Recurrent High Grade Glioma	1/2	41	HSPPC-96 vaccine	OS at 6 mo = 90.2% (95% CI 75.9–96.8%) mOS = 42.6 weeks (95% CI = 34.7–50.5)	NCT00293423
	A Phase II Randomized Trial Comparing the Efficacy of Heat Shock Protein Peptide Complex-96 (HSPPC-96) Vaccine Given with Bevacizumab versus Bevacizumab Alone in the Treatment of Surgically Resectable Recurrent Glioblastoma	2	90	HSPPC-96 vaccine + bevacicumab vs. Bevacizumab alone	OS = 7.5 vs. 10.7 mo (HR = 2.06)	NCT01814813

TMZ = temozolomide; DC = dendritic cell; GBM = glioblastoma multiforme; HSP = heat shock protein; mOS = median overall survival; mPFS = median progression-free survival; CI = confidence interval. * Secondary endpoints.

**Table 3 cancers-14-04023-t003:** Immune checkpoint inhibitor clinical trials for GBM.

	Study Title	Phase	Size	Intervention Details	Primary Endpoint	Reference
Newly diagnosed GBM	An Investigational Immuno-therapy Study of Nivolumab Compared to Temozolomide, Each Given With Radiation Therapy, for Newly-diagnosed Patients With Glioblastoma (GBM, a Malignant Brain Cancer) (Checkmate 498)	3	560	Nivolumab + RT vs. TMZ + RT	OS (endpoint not met) mOS = 13.4 vs. 14.9 mo (HR = 1.31, 95% CI = 1.09–1.58, *p* = 0.0037)	NCT02617589
	An Investigational Immuno-therapy Study of Temozolomide Plus Radiation Therapy With Nivolumab or Placebo, for Newly Diagnosed Patients With Glioblastoma (GBM, a Malignant Brain Cancer) (Checkmate 548)	3	716	Nivolumab + RT + TMZ vs. Placebo + RT + TMZ	PFS = 10.6 vs. 10.3 mo (HR = 1.06) mOS = 28.9 vs. 32.1 mo (HR = 1.10)	NCT02667587
	Phase I/II Study to Evaluate the Safety and Clinical Efficacy of Atezolizumab (aPDL1) in Combination With Temozolomide and Radiation in Patients With Newly Diagnosed Glioblastoma (GBM)	1/2	60	Atezolizumab + RT + TMZ vs. adjuvant atezolizumab + TMZ	MTD, mOS = 17.1 mo	NCT03174197
Recurrent GBM	A Study of the Effectiveness and Safety of Nivolumab Compared to Bevacizumab and of Nivolumab With or Without Ipilimumab in Glioblastoma Patients (Checkmate 143)	3	369	Nivolumab vs. Bevacizumab	mOS = 9.8 vs. 10.0 mo (HR = 1.04, 95% CI = 0.83–1.30)	NCT02017717
	Study of Pembrolizumab (MK-3475) in Participants With Advanced Solid Tumors (MK-3475-028/KEYNOTE-28)	1	26	Pembrolizumab	mOS = 13.1 mo	NCT02054806
	Pembrolizumab +/− Bevacizumab for Recurrent GBM	2	80	Pembrolizumab + bevacizumab vs. Pembrolizumab	PFS at 6 mo = 26.0% (95% CI 16.3–41.5) vs. 6.7% (95% CI 1.7–25.4)	NCT02337491
	A Study of Atezolizumab (an Engineered Anti-Programmed Death-Ligand 1 [PDL1] Antibody) to Evaluate Safety, Tolerability and Pharmacokinetics in Participants With Locally Advanced or Metastatic Solid Tumors	1	16	Atezolizumab	Safety profile, mOS = 4.2 mo	NCT01375842

GBM = glioblastoma multiforme; RT = radiation therapy; TMZ = temozolomide; PDL1 = programmed death ligand 1; OS = overall survival; mOS = median overall survival; PFS = progression-free survival; HR = hazard ratio; CI = confidence interval.

**Table 4 cancers-14-04023-t004:** Treg and T-cell activator clinical trials for GBM.

	Target	Study Title	Phase	Size	Intervention	Primary Outcome	Reference
TReg Targeting	IDO1	A Phase 1/2a Study of BMS-986205 Administered in Combination With Nivolumab (Anti-PD-1 Monoclonal Antibody) and in Combination With Both Nivolumab and Ipilimumab (Anti-CTLA-4 Monoclonal Antibody) in Advanced Malignant Tumors	1	30	IDO1 inhibitor (BMS 986205) + RT + nivolumab with vs. without TMZ	MTD	NCT04047706
	GITR	A Phase II Study of the Anti-GITR Agonist INCAGN1876 and the PD-1 Inhibitor INCMGA00012 in Combination With Stereotactic Radiosurgery in Recurrent Glioblastoma	2	32	Anti-GITR agonist (INCAGN1876) + PD1 inhibitor (INCMGA00012) + SRS with vs. without surgery	Objective radiographic response	NCT04225039
T-Cell Activating	41BB	A Phase 1b study of utomilumab (PF-05082566) in combination with mogamulizumab in patients with advanced solid tumors	1/1b	24	Utomilumab (PF-05082566) + CCR4 mAb	MTD	NCT02444793
	OX-40L	Phase I Trial of DNX-2440 Oncolytic Adenovirus in Patients With Recurrent Glioblastoma	1	24	DNX-2440 virus	MTD	NCT03714334
	CD40L	A Phase 1 Trial of D2C7-IT in Combination With an Fc-engineered Anti-CD40 Monoclonal Antibody (2141-V11) Administered Intratumorally Via Convection-Enhanced Delivery for Adult Patients With Recurrent Malignant Glioma	1	8	CD40 agonist antibody (2141-V11) + EGFRvIII immunotoxin (D2C7-IT)	MTD	NCT04547777

Treg = regulatory T cell; IDO1 = indoleamine 2,3 dioxygenase 1; CTLA-4 = cytotoxic T-lymphocyte-associated protein 4; RT = radiation therapy; TMZ = temozolomide; GITR = clucocorticoid-induced TNFR-related protein; PD-1 = programmed death-1; SRS = stereotactic radiosurgery; CCR4 = C-C chemokine receptor type 4; CD40L = cluster of differentiation 40 ligand; EGFRvIII = epidermal growth factor receptor variant III; MTD = maximum tolerated dose.

**Table 6 cancers-14-04023-t006:** Clinical trials aimed at enhancing immunotheraeutic response in GBM.

Primary Mechanism	Study Title	Phase	Size	Intervention	Primary Outcome	Reference
STING agonism	Intratumoral Delivery of STING Agonist Results in Clinical Responses in Canine Glioblastoma	Preclincial	6	STING agonist IACS-8779	MTD and radiographic response	N/A
STAT3 inhibition	A first-in-human Phase I trial of the oral *p*-STAT3 inhibitor WP1066 in patients with recurrent malignant glioma	1	8	*p*-STAT3 inhibitor WP1066	MTD. * mOS = 25 mo.	NCT02977780
IL-12 therapy	An Open-Label, Multi-Center Trial of INO-5401 and INO-9012 Delivered by Electroporation (EP) in Combination With REGN2810 in Subjects With Newly-Diagnosed Glioblastoma (GBM)	1/2	52	INO-9012 (IL-12 plasmid) and INO-5401 + cemiplimab + TMZ + radiation	Safety (ongoing)	NCT03491683
	A Phase I Study of Ad-RTS-hIL-12, an Inducible Adenoviral Vector Engineered to Express hIL-12 in the Presence of the Activator Ligand Veledimex in Subjects With Recurrent or Progressive Glioblastoma or Grade III Malignant Glioma	1	31	Veledimex + Ad-RTS-hIL-12	Safety and tolerability. * mOS = 12.7 mo	NCT02026271
BBB disruption	A Study to Evaluate the Safety of Transient Opening of the Blood-Brain Barrier by Low Intensity Pulsed Ultrasound With the SonoCloud Implantable Device in Patients With Recurrent Glioblastoma Before Chemotherapy Administration	1/2	21	SonoCloud + carboplatin	Safety. * mPFS = 4.11 mo, mOS = 12.94 mo	NCT02253212
	A Study to Evaluate the Safety and Feasibility of Blood-Brain Barrier Disruption Using Transcranial MRI-Guided Focused Ultrasound With Intravenous Ultrasound Contrast Agents in the Treatment of Brain Tumours With Doxorubicin	1	5	ExABlate	Safety	NCT02343991
	A Study to Evaluate the Safety and the Efficacy of Transient Opening of the Blood-brain Barrier (BBB) by Low Intensity Pulsed Ultrasound With the SonoCloud-9 Implantable Device in Recurrent Glioblastoma Patients Eligible for Surgery and for Carboplatin Chemotherapy	1/2a	33	Sonocloud-9 device + carboplatin	MTD and BBB opening (ongoing)	NCT03744026
	Phase 1/2 Trial of Blood-brain Barrier Opening With an Implantable Ultrasound Device SonoCloud-9 and Treatment With Albumin-bound Paclitaxel in Patients With Recurrent Glioblastoma	1/2	17	Sonocloud-9 device + paclitaxel + carboplatin	Maximum tolerated dose and 1-year survival rate (ongoing)	NCT04528680

STING = stimulator of interferon genes; STAT3 = signal transducers and activators of transcription 3; IL-12 = interleukin-12; BBB = blood–brain barrier; GBM = glioblastoma multiforme; TMZ = temozolomide; MTD = maximum tolerated dose; mOS = median overall survival; mPFS = median progression-free survival. * = secondary endpoint.

**Table 7 cancers-14-04023-t007:** Immunotherapy clinical trials aimed at the innate immune system for GBM.

Target	Study Title	Phase	Size	Intervention	Primary Outcome	Reference
MDSC	Targeting Myeloid Derived Suppressor Cells in Recurrent Glioblastoma: Phase 0/1 Trial of Low Dose Capecitabine + Bevacizumab in Patients With Recurrent Glioblastoma	0/1	4	Capecitabine + bevacizumab	MDSC reduction ranging between 20% and 79% from baseline	NCT02669173
	A Phase I/II, Open-label, Multi-center Study of the Safety and Efficacy of BLZ945 as Single Agent and in Combination With PDR001 in Adults Patients With Advanced Solid Tumors	1/2	146	CSF1R inhibitor (BLZ945) with vs. without PD1 blockade (PDR001)	MTD, 6 mo PFS	NCT02829723
	A Phase 2 Study of Orally Administered PLX3397 in Patients With Recurrent Glioblastoma	2	38	CSF1R inhibitor (PLX3397)	6 mo PFS = 8.6%	NCT01349036
Niacin (ongoing)	A Phase I-II Study of Niacin in Patients With Newly Diagnosed Glioblastoma Receiving Concurrent Radiotherapy and Temozolomide Followed by Monthly Temozolomide	1/2	59	Niacin + RT + TMZ	MTD, 6mo PFS	NCT04677049
Gamma delta cells (ongoing)	A Phase I Study of Drug Resistant Immunotherapy (DRI) With Activated, Gene Modified γδ T Cells in Patients With Newly Diagnosed Glioblastoma Multiforme Receiving Maintenance Temozolomide Chemotherapy	1	12	Gene-modified gamma delta T cells	MTD	NCT04165941

MDSC = myeloid-derived suppressor cells; PD1 = programmed death 1; CSF1R = colony-stimulating factor 1 receptor; RT = radiation therapy; TMZ = temozolomide; PFS = progression-free survival; MTD = maximum tolerated dose.

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
