# Peer review of "Next Steps for Immunotherapy in Glioblastoma"

_cancers, 2022, doi:10.3390/cancers14164023_

Round 1
Reviewer 1 Report
This is a well-written review on immunotherapy in glioblastoma. However, it is relatively condensed with numerous topics being discussed in one manuscript. In order to make the manuscript more reader-friendly, authors should consider:
1) Including illustrative images for some of the subsections to summarize the specific therapeutic approaches.
2) Creating more tables that summarize some of the sections.
Other points to consider:
1) Intratumoural heterogeneity in glioblastoma is a well-established phenomenon that is thought to be one of the main factors in the failure of several treatment strategies. Authors should consider discussing this issue.
2) In 4.3, the authors mentioned that PTEN mutations appear to be associated with reduced response to anti-PD-1 therapy. This finding is open to discussion, as the majority of glioblastomas undergo chr10q deletion, which harbours the PTEN gene. These tumours do not necessarily carry PTEN mutations; however, they do have a loss of PTEN function. Therefore, caution is warranted when creating an association between PTEN mutation and treatment success.
3) Authors discussed ultrasound to disrupt the BBB. However, irradiation has been shown as a very effective method in achieving such a goal, particularly when the patients are already receiving irradiation therapy. Authors should consider including low IR treatment in disrupting BBB.
Author Response
Reviewer 1 Comments
This is a well-written review on immunotherapy in glioblastoma. However, it is relatively condensed with numerous topics being discussed in one manuscript. In order to make the manuscript more reader-friendly, authors should consider:
- Including illustrative images for some of the subsections to summarize the specific therapeutic approaches.
Response: We thank the reviewer for this comment. We have now provided several figures in sections 2.1.1, 2.2, 2.3, 5.3, and 6.1 to help illustrate some points.
- Creating more tables that summarize some of the sections.
Response: We thank the reviewer for this feedback and agree that summarizing the information in table format would help make the manuscript more reader-friendly. We have included 6 additional tables highlighting the various clinical trials that are reviewed in this article.
Other points to consider:
- Intratumoural heterogeneity in glioblastoma is a well-established phenomenon that is thought to be one of the main factors in the failure of several treatment strategies. Authors should consider discussing this issue.
Response: We have reviewed the role of intratumoral heterogeneity in treatment response in section 3 on page 14.
- In 4.3, the authors mentioned that PTEN mutations appear to be associated with reduced response to anti-PD-1 therapy. This finding is open to discussion, as the majority of glioblastomas undergo chr10q deletion, which harbours the PTEN gene. These tumours do not necessarily carry PTEN mutations; however, they do have a loss of PTEN function. Therefore, caution is warranted when creating an association between PTEN mutation and treatment success.
Response: We appreciate the reviewer’s suggestion. We have now mentioned that chromosome 10q deletion occurs in glioblastoma and results in PTEN loss in section 4.3 on pages 15-16.
- Authors discussed ultrasound to disrupt the BBB. However, irradiation has been shown as a
very effective method in achieving such a goal, particularly when the patients are already receiving irradiation therapy. Authors should consider including low IR treatment in disrupting BBB.
Response: We have now included the role of low irradiation therapy in BBB disruption in section 5.5 on page 19-20.

Reviewer 2 Report
This excellent and well-written manuscript reviews the current status and promising future directions for immunotherapy of GBM (excluding CAR-T cell therapy). There is good attention to potential factors modifying the GBM response to immunotherapy. The review is fairly thorough overall and will be a useful addition to the literature, but there are definitely some additional components that should be added.
--In the sub-section on rindopepimut trials, it is worth mentioning that it was found that the subsequent GBM was found to have largely lost EGFRvIII expression—this loss of targeted antigen, and/or outgrowth of target-negative cells, has important implications for a number of immunotherapies targeting GBM and should be discussed (beyond the passing mention of antigen escape in the subsequent paragraph on multi-peptide vaccines).
--Mention of vaccines with the IDH1 mutant peptide should be added.
--A bit more on CeGaT would be helpful, including a reference.
--It is likely worth discussing the issues with historical controls used in many of these immunotherapy studies.
--The paragraph on why checkpoint blockade has been unsuccessful should be bolstered.
--I would definitely add a paragraph about the potential benefit of neoadjuvant anti-PD1 immunotherapy—many of us believe this is promising (and it may be worth mentioning that it also looks more promising in other cancers as well, and there are references to support this).
-- The section on MDSCs should reference the excellent recent paper on gender differences in MDSCs in GBM.
--The section on therapies to redirect microglia/macrophages against GBM should include other work as well, including the good recent paper on high-dose niacin (which now seems to be in a clinical trial in Canada).
--I believe a section on oncolytic viral therapies should be added, given that their principal mechanism is likely to be immunotherapeutic.
Author Response
Reviewer 2 Comments
This excellent and well-written manuscript reviews the current status and promising future directions for immunotherapy of GBM (excluding CAR-T cell therapy). There is good attention to potential factors modifying the GBM response to immunotherapy. The review is fairly thorough overall and will be a useful addition to the literature, but there are definitely some additional components that should be added.
In the sub-section on rindopepimut trials, it is worth mentioning that it was found that the subsequent GBM was found to have largely lost EGFRvIII expression—this loss of targeted antigen, and/or outgrowth of target-negative cells, has important implications for a number of immunotherapies targeting GBM and should be discussed (beyond the passing mention of antigen escape in the subsequent paragraph on multi-peptide vaccines).
Response: We thank the reviewer for this important feedback. We have now noted the subsequent loss of EGFRvIII expression in the rindopepimut trials in section 2.1.1 on page 2. However, we have also noted that EGFR loss has been shown to occur regardless of treatment.
Mention of vaccines with the IDH1 mutant peptide should be added.
Response: We have now added a section on the IDH1 mutant peptide in section 2.1.1 on pages 3.
A bit more on CeGaT would be helpful, including a reference.
Response: We thank the reviewer for this point. To our knowledge, no preclinical or clinical trials have been published on the CeGaT vaccine.
It is likely worth discussing the issues with historical controls used in many of these immunotherapy studies.
Response: We have included a section discussing the use of historical controls in many of these trials in section 2.1.2 on page 7.
The paragraph on why checkpoint blockade has been unsuccessful should be bolstered.
Response: We thank the reviewer for this feedback. We have added some additional information on this subject in section 2.2 on page 11.
I would definitely add a paragraph about the potential benefit of neoadjuvant anti-PD1 immunotherapy—many of us believe this is promising (and it may be worth mentioning that it also looks more promising in other cancers as well, and there are references to support this).
Response: We thank the reviewer for this comment and agree that these exciting trials should be included for a comprehensive understanding of current immunotherapy research. We have included a paragraph on neoadjuvant anti-PD1 immunotherapy in section 2.2 on page 11.
The section on MDSCs should reference the excellent recent paper on gender differences in MDSCs in GBM.
Response: We have now included this interesting paper on the effect of gender on MDSCs in GBM in section 6.1 on page 22.
The section on therapies to redirect microglia/macrophages against GBM should include other work as well, including the good recent paper on high-dose niacin (which now seems to be in a clinical trial in Canada).
Response: We appreciate the reviewer’s suggestion and for highlighting this important new work. We have included a discussion on high dose niacin in GBM in section 6.1 on page 23.
I believe a section on oncolytic viral therapies should be added, given that their principal mechanism is likely to be immunotherapeutic.
Response: We have added a section on oncolytic viral therapies in section 5.1 on page 17 and referenced a separate review article for more detailed discussion.

Reviewer 3 Report
Authors have a lot of important information about glioblastoma immunotherapy to share but they need to reduce the reader's burden by putting results of all completed studies in tables. The tables should be able to be read independently of the text (acronyms and abbreviations must be defined in footnotes). Studies in progress should be included in separate tables.
Author Response
Reviewer 3 Comments
Authors have a lot of important information about glioblastoma immunotherapy to share but they need to reduce the reader's burden by putting results of all completed studies in tables. The tables should be able to be read independently of the text (acronyms and abbreviations must be defined in footnotes). Studies in progress should be included in separate tables.
Response: We thank the reviewer for this feedback, and agree that summarizing completed and ongoing studies in tables would help make the manuscript more reader-friendly. We have added 6 additional tables and hope that these changes are acceptable.
Round 2
Reviewer 3 Report
Provides a comprehensive list of immunotherapeutic treatments. Most recent studies are of most interest and should therefore be discussed in the most depth.
See attached for minor comments.
